# High-density vertical sidewall MoS₂ transistors through T-shape vertical lamination

Quanyang Tao[1,2,4], Ruixia Wu[1,3,4], Xuming Zou ⬤[1] ✉, Yang Chen[1], Wanying Li[1], Zheyi Lu[1], Likuan Ma[1], Lingan Kong ⬤[1], Donglin Lu[1], Xiaokun Yang[1], Wenjing Song[1], Wei Li[3], Liting Liu[1], Shuimei Ding[1], Xiao Liu[1], Xidong Duan ⬤[3], Lei Liao ⬤[2] ✉ & Yuan Liu ⬤[1] ✉

Vertical transistors, in which the source and drain are aligned vertically and the current flow is normal to the wafer surface, have attracted considerable attention recently. However, the realization of high-density vertical transistors is challenging, and could be largely attributed to the incompatibility between vertical structures and conventional lateral fabrication processes. Here we report a T-shape lamination approach for realizing high-density vertical sidewall transistors, where lateral transistors could be pre-fabricated on planar substrates first and then laminated onto vertical substrates using T-shape stamps, hence overcoming the incompatibility between planar processes and vertical structures. Based on this technique, we vertically stacked 60 MoS₂ transistors within a small vertical footprint, corresponding to a device density over $10^8$ cm$^{-2}$. Furthermore, we demonstrate two approaches for scalable fabrication of vertical sidewall transistor arrays, including simultaneous lamination onto multiple vertical substrates, as well as on the same vertical substrate using multi-cycle layer-by-layer laminations.

Field-effect transistors (FETs) represent the fundamental building blocks for modern computer processors. Today's dominant device architecture is lateral-transport FET (ref. [1,2]), which layers transistors along a wafer's surface. By making the lateral size of transistors smaller and smaller through advanced lithography, more transistors can be integrated into a single chip with much improved integration density[3]. Today, as transistor pitch size (lateral distance between two neighbor transistors) enters the sub-50-nm regime with increasing technical challenges[4], the exploration of alternative device geometries is ever more important for further increasing the transistor density and extending Moore's law. Vertical-transport FETs (VFETs)[5-9], on the other hand, layer transistors perpendicularly to the wafer plane and direct the current flow vertically to the wafer surface. Within this device structure, each device is stacked on top of another and does not consume additional chip footprint beyond what is needed for a single device placed at the bottom[10-13], opening up another dimension for high-density integrated circuits.

However, fabricating vertical transistors is technologically challenging, and could be largely attributed to the incompatibility with conventional lateral-based fabrication processes. For example, to fabricate a single vertical transistor, the bottom source electrode layer needs to be created first, followed by the deposition of the channel layer and the drain electrode layer. To stack multiple devices within the vertical direction, these processes have to be repeated, leading to

---

¹Key Laboratory for Micro-Nano Optoelectronic Devices of Ministry of Education, School of Physics and Electronics, Hunan University, Changsha, China. ²Changsha Semiconductor Technology and Application Innovation Research Institute, College of Semiconductors (College of Integrated Circuits), Hunan University, Changsha, China. ³State Key Laboratory for Chemo/Biosensing and Chemometrics, College of Chemistry and Chemical Engineering, Hunan University, Changsha, China. ⁴These authors contributed equally: Quanyang Tao, Ruixia Wu. ✉e-mail: zouxuming@hnu.edu.cn; liaolei@whu.edu.cn; yuanliuhnu@hnu.edu.cn

complex fabrication processes with low throughput. Hence, the state-of-the-art vertical transistor is largely limited to the demonstration of a few device layers in vertical direction[5,14–18], limiting the achievement of high-density devices. This is in great contrast to the planar integrated circuit techniques based on advanced lithography and deposition processes, where large amounts of lateral devices (over millions or billions) could be batch-fabricated in parallel. These differences originate from the fact that conventional lateral processes are top-down approaches involving physical particles such as photons (in lithography), reactive ions (in etching), or physical/chemical vapors (in deposition), which can only generate multiple structures within the wafer plane. Creating scalable structures vertically (perpendicular to the wafer plane) using these techniques is difficult since these physical particles (photons, ions, vapors) can not be applied to the vertical substrate. Therefore, developing a new process that can generate multiple structures vertically is critical for the realization of high-density VFETs.

In this work, we report an alternative approach to fabricating multiple vertical sidewall transistors simultaneously. Lateral transistors could be pre-fabricated on a planar substrate using conventional batch processes, further dry-released, and laminated onto the vertical substrate through a custom-designed T-shape stamp, hence overcoming the incompatibility between planar processes and vertical structures. Importantly, owing to the low strain induced during the dry-lamination process, the transistors could conformally contact with the vertical substrate without damage or degradation, leading to high-performance vertical transistors with low device-to-device variation. Based on this technique, we vertically stacked 60 $MoS_2$ transistors within a small footprint of 0.035 $\mu m^2$, corresponding to a theoretical device density of $1.7 \times 10^{11}$ cm$^{-2}$ and $2.4 \times 10^8$ cm$^{-2}$, depending on the definition of device area. Finally, we demonstrate two approaches for scalable fabrication of vertical sidewall transistor arrays, including simultaneous lamination onto multiple vertical substrates, as well as multi-cycle layer-by-layer laminations on the same vertical substrate. Our results offer an alternative route for the vertical electronics and high-density transistors.

## Results

### Vertical lamination processes using T-shape stamp

Figure 1a–e schematically illustrates the vertical lamination processes of two-dimensional $MoS_2$ transistors. First of all, lateral $MoS_2$ transistors are pre-fabricated on a silicon sacrifice wafer, using conventional planar processes of lithography and thermal deposition. The whole device layer (both $MoS_2$ channel and Au electrodes) could be mechanically released from the sacrificial substrate with the assistant of PMMA (polymethyl methacrylate) capping layer using our previously developed method[19–21] (detailed in "Methods"), as shown in Fig. 1a. We note the dry releasing approach is essential to avoid the involvement of solutions, hence maintaining the intrinsic properties of $MoS_2$ channel and metal contact. Next, the vertical

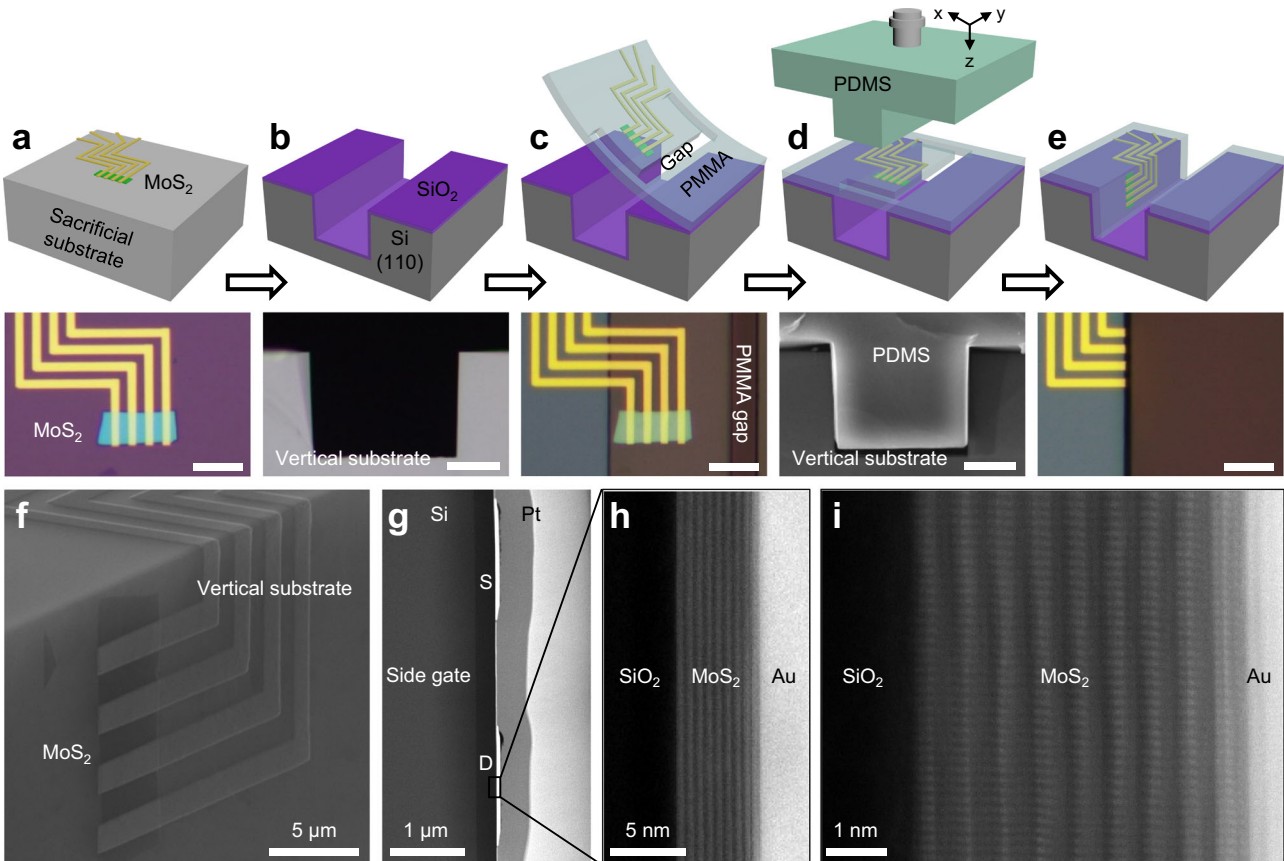

**Fig. 1 | Vertical lamination processes and characterization of $MoS_2$ based vertical sidewall transistors. a–e** Schematics and the corresponding optical images of the vertical lamination processes with 5 steps: planar $MoS_2$ transistors pre-fabricated on a sacrificial substrate (**a**), fabrication of vertical silicon trench through etching (**b**), dry-transfer of $MoS_2$ transistors with the polymethyl methacrylate (PMMA) gap on top of Si trench (**c**), T-shape polydimethylsiloxane (PDMS) stamp laminated and pushed into the trench (**d**), and $MoS_2$ vertical transistors after lamination (**e**). **f** Scanning electron microscopy (SEM) image of the fabricated $MoS_2$ vertical transistors on vertical substrate. **g–i** Cross-sectional SEM image (**g**) and scanning transmission electron microscopy (STEM) images (**h, i**) of the vertical transistors, indicating the optimized interface after vertical lamination. Scale bars are 10 $\mu m$ for panel **a, c, e**, 20 $\mu m$ for panel **b, d**. S, source; D, drain.

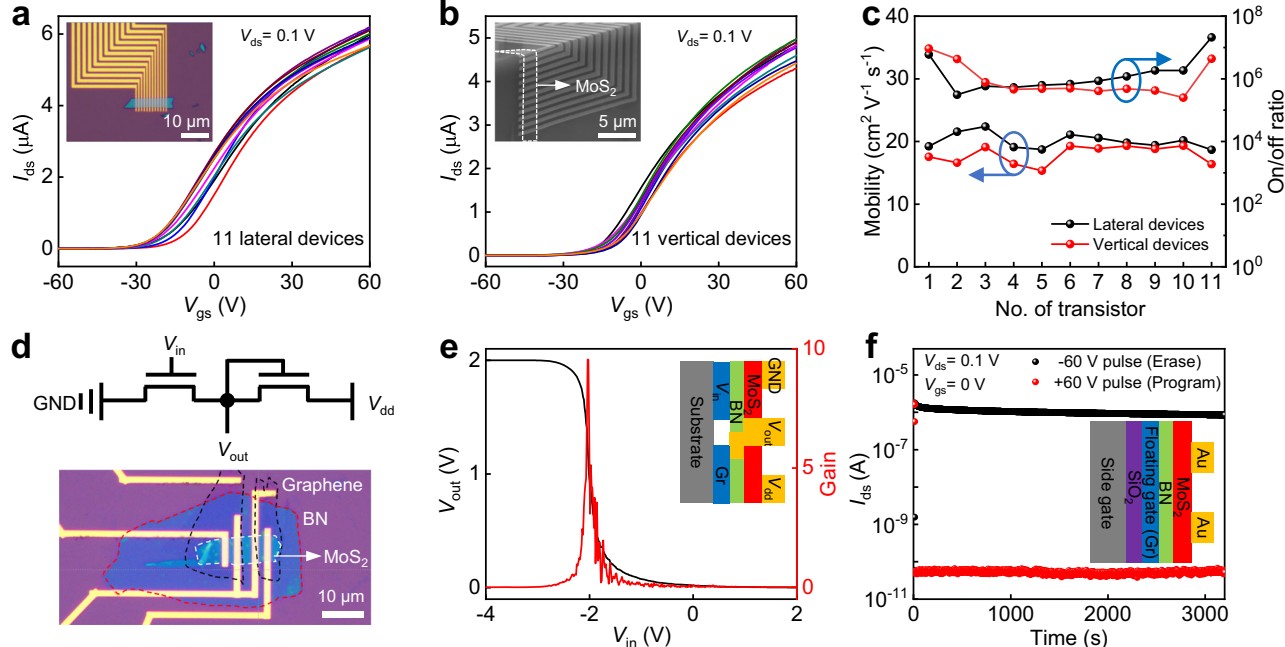

**Fig. 2 | Electrical characterizations of MoS$_2$ transistors. a, b** Transfer characteristics of the 11 parallel MoS$_2$ transistors on the pre-fabricated lateral substrate (**a**) as well as after lamination on the vertical substrate (**b**). Insets are the corresponding device images. **c** Comparison of the mobilities and on/off ratios for lateral and vertical transistors, where comparable electrical properties and low device-to-device variations are observed. **d** Circuit diagram (upper) and optical image of pre-fabricated lateral NMOS inverter composed of two MoS$_2$ transistors in series. The black, white, and red dashed boxes are graphene, MoS$_2$, and BN, respectively. **e** The voltage transfer characteristic (black line) and voltage gain (red line) of the vertical inverter at supply voltage of 2 V. **f** Retention performance of the vertical floating gate memory. Insets are the schematics of the inverter (**e**) and memory (**f**), respectively. $V_{gs}$, gate-source voltage; $V_{ds}$, drain-source voltage; $I_{ds}$, drain-source current; GND, ground; $V_{in}$, input voltage; $V_{out}$, output voltage; $V_{dd}$, supply voltage.

substrate is prepared by wet etching of Si substrate[22–24] ("Methods"), creating a deep Si sidewall (~40 μm depth), as shown in Fig. 1b. Furthermore, the released device layer is laminated on top of the Si vertical substrate (covered by 300 nm thick vertical SiO$_2$) using dry-alignment transfer process, as schematically illustrated in Fig. 1c and Supplementary Fig. 1. In conventional lamination processes, the laminated device layer can only form good contact with the bottom flat surface, the integration of the devices on the deep vertical substrate could lead to the suspension of device layer on the silicon sidewall, as shown in Fig. 1c. To overcome this challenge, a T-shape PDMS (polydimethylsiloxane) stamp is designed (Methods), where the suspended MoS$_2$ transistors could be precisely pushed onto the vertical substrate, leading to the intimate contact between the device layer and the vertical substrate (Fig. 1d, e).

We note our T-shape vertical lamination process is unique to achieve vertically stacked transistors, owing to two factors. First of all, the T-shape PDMS stamp is fabricated using the same silicon vertical trench as the mold (Methods), hence, the stamp has the exact same structure to fit the deep silicon trench, as shown in Fig. 1d. When mechanically moving the micro-sized T-shape stamp into the trench, it could push the device layer onto the vertical sidewall to form intimate contact between the device layer and the vertical substate (Fig. 1e). This can be confirmed in the scanning electron microscopy (SEM) image as well as the scanning transmission electron microscopy (STEM) images (Fig. 1f–i), where the device layer is closely contacted with the vertical substrate without any air bubbles. Second, both the device releasing process and the vertical lamination process are based on dry processes without solvents and mechanical stress. In contrast, the conventional wet-transfer process[25] on the non-flat substrate (or vertical substrate) is based on capillary force between the substrate during the solution evaporation, leading to a large stretching force and the distortion of the device layer, as shown in our control experiment in Supplementary Fig. 2.

## Electrical properties of vertical sidewall MoS$_2$ transistors

To demonstrate the robustness of our T-shape vertical lamination technique, we have measured and compared the electrical properties of the MoS$_2$ transistors on both lateral substrate and vertical substrate. As shown in Fig. 2a, a lateral MoS$_2$ transistor array is fabricated on a conventional flat substate, where highly doped Si substrate is used as the back gate, 300 nm thick SiO$_2$ is used as the gate dielectric, Au (50 nm thick) is used as the source/drain electrodes, respectively. The electrical measurements of the as-fabricated lateral devices were conducted at room temperature in a vacuum probe station (10$^{-4}$ Torr). In general, n-type transfer behaviors (Fig. 2a) are observed with an on-off ratio over 10$^6$, demonstrating decent electrical properties and consistent with previous literatures[20,26,27]. In particular, the 11 parallel devices demonstrate low device-to-device variation in terms of on-state current (<10%), indicating the optimized device conditions using our fabrication techniques. After electrical measurements, these planar transistors are released from flat back-gate substrate and further laminated on the vertical substrate using our T-shape lamination process described in Fig. 1. We note the sidewall of highly doped silicon is also covered with 300 nm thick SiO$_2$, hence, global side gate with identical dielectric conditions is achieved. As shown in Fig. 2b, the vertical transistors demonstrate similar n-type switching behaviors, indicating the T-shape lamination process won't alter the intrinsic properties of the MoS$_2$ channel and contact conditions. Figure 2c summarizes the on-off ratios and field effect mobilities of these devices on both lateral and vertical substrates, where similar electrical properties are observed, confirming the device layer is conformally contacted to the vertical side gate with efficient gate modulation. To further investigate the device uniformity, we have fabricated 60 planar and vertical transistors, and statistically analyzed their transfer curves. As shown in Supplementary Fig. 3, their key electrical properties demonstrate low device-to-device variations, further confirming the transistors remain free from damage or degradation after vertical

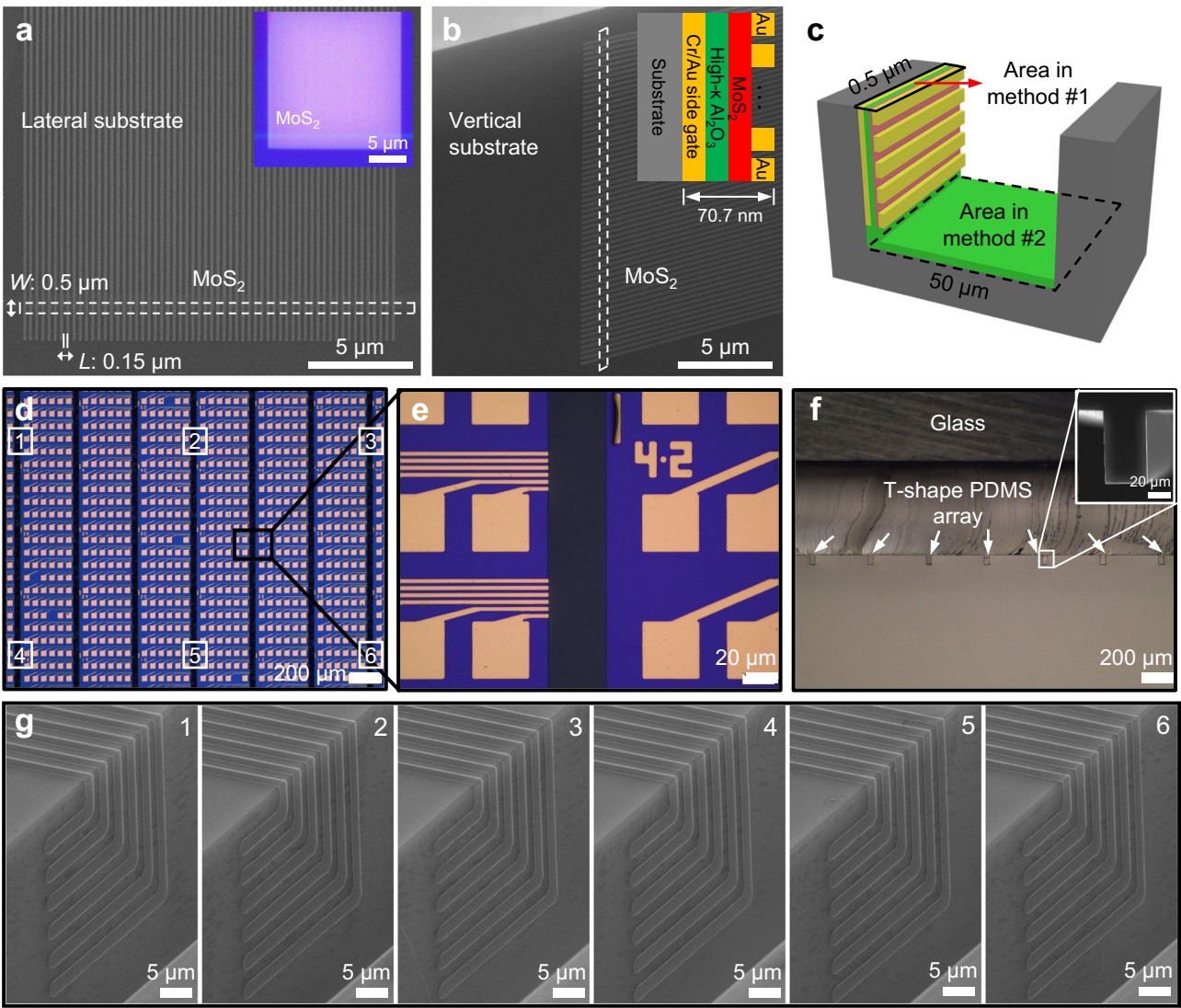

**Fig. 3 | Scalable fabrication of vertical sidewall transistors. a** 60 lateral mono-layer MoS$_2$ transistor with 0.15 μm channel length (*L*) and 0.5 μm channel width (*W*). Inset is the optical image of lateral MoS$_2$ transistors. **b** SEM image of vertical MoS$_2$ transistors. Inset is the corresponding cross-sectional schematic. The white dashed boxes are MoS$_2$ in panels **a**, **b**. **c** Schematic of the definition of device area in the density calculation. **d** Optical image of vertical device arrays with a pitch size of 420 μm in the *x*-direction and 90 μm in the *y*-direction. **e** Zoomed-in optical image of vertical device arrays. **f** Cross-sectional image of T-shape PDMS stamp array, inset is the cross-sectional SEM image of one T-shape PDMS stamp. The white arrows point to T-shape PDMS stamps. **g** SEM images of vertical devices at different locations (marked in (**d**)).

lamination. Besides global side-gate devices, we could also fabricate separated gates for vertical MoS$_2$ devices using 10 nm Al$_2$O$_3$ as the dielectric, as shown in Supplementary Fig. 4.

Furthermore, our T-shape vertical lamination technique could be well extended to more complex device structures. To demonstrate this, a MoS$_2$-based NMOS inverter is pre-fabricated by connecting two transistors in series, as schematically illustrated in Fig. 2d. After integrating the whole device layer on the vertical substrate (Fig. 2e, inset and Supplementary Fig. 5a), the inverter demonstrates sharp output voltage (*V*$_{out}$) transition as a function of input voltage (*V*$_{in}$), yielding a voltage gain (defined as |d*V*$_{out}$/d*V*$_{in}$|) of 10 at supply voltage (*V*$_{dd}$) of 2 V (Fig. 2e). Beyond logic devices, our technique could also be applied to fabricated other vertical devices such as memory. For example, a four-layer van der Waals heterostructure (graphene/BN/MoS$_2$/Au) memory[28] could also be pre-fabricated on a lateral substrate, where the graphene, BN, MoS$_2$, and Au are used as a floating gate, tunneling dielectric, semiconductor channel, and metal contact, respectively. After laminating the memory using a T-shape PDMS stamp, the vertical device demonstrates a large memory window using a side control gate,

and the program and erase states show a significant program/erase (P/E) current ratio exceeding 10$^4$ (Supplementary Fig. 5b). Importantly, the large P/E ratio remains after more than 3000 s measurement (Fig. 2f), indicating the high durability of our vertical memory.

## Scalable fabrication of high-density transistors

Achieving high integration density is the primary motivation for stacking vertical transistors, where multiple devices could be stacked together in the vertical direction and do not consume additional chip footprint beyond what is needed for a single device placed at the bottom. To achieve high-density vertical devices, we have pre-fabricated 60 short channel MoS$_2$ transistors on a sacrificial planar wafer, and the channel length, contact length, and channel width are 0.15, 0.15 and 0.5 μm, respectively, as shown in Fig. 3a. Furthermore, we have laminated these short-channel devices on the vertical substrate using our vertical lamination technique, where the side gate is 30 nm thick Cr/Au, the side dielectric is 10 nm thick Al$_2$O$_3$, the monolayer MoS$_2$ is 0.7 nm, and the source/drain electrodes are 30 nm thick Au, as shown in Fig. 3b. The electrical

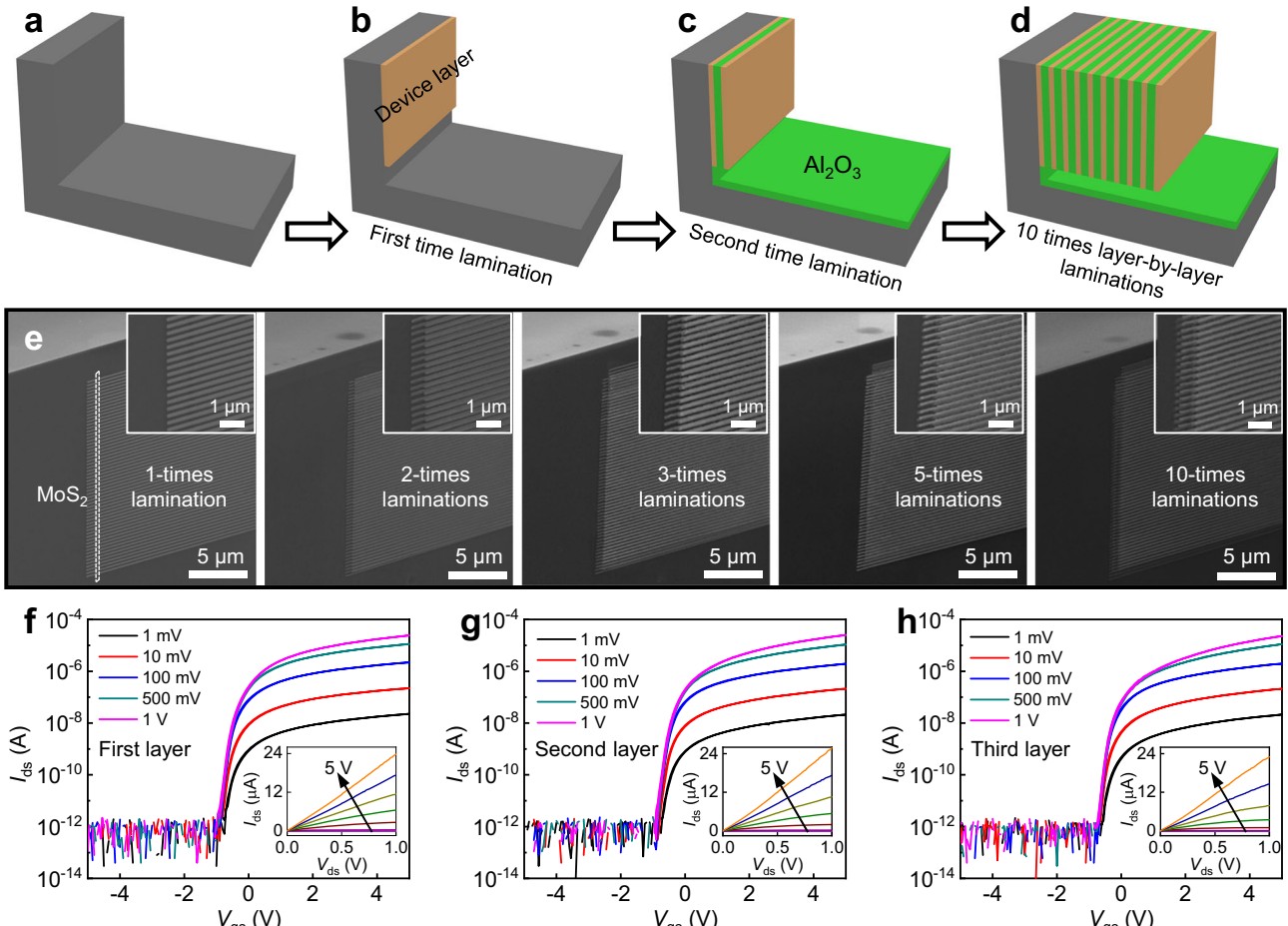

**Fig. 4 | Layer-by-layer vertical integrations of high-density vertical devices. a–d** Schematics of the 10 times layer-by-layer vertical laminations of vertical transistors on the same sidewall. **e** The corresponding SEM images of vertical devices. The channel length and contact length of all devices are fixed at 0.15 μm, the channel width is 0.2 μm, the source/drain electrodes are 20 nm thick Au, and the interlayer dielectric (between two adjacent vertical layers) is 10 nm thick $Al_2O_3$.

Insets are the corresponding zoomed-in SEM images of vertical devices. **f–h** Transfer curves and output curves (inset) of the first-layer (**f**), second-layer (**g**), and third-layer (**h**) $MoS_2$ transistor, demonstrating consistent electrical properties. For output curves, various gate voltages from − 5 V to 5 V are applied with a step of 1 V.

measurements of short-channel vertical devices are shown in Supplementary Fig. 6, exhibiting on-state current of up to 400 μA/μm, small subthreshold swing of 110 mV/dec and high on-off ratio of $10^9$.

The device density could be further calculated using the device number divided by the total device area. Here, the device number is 60, and the device area is defined using two different methods. In the first method (method #1), the total device area is the planar region of one device, which is ~ 0.035 μm² (70.7 nm × 0.5 μm) including all active components (gate electrode, dielectric layer, $MoS_2$ channel, and source/drain electrodes), as schematically illustrated in Fig. 3c. Therefore, the highest density of $1.7 \times 10^{11}$ cm$^{-2}$ could be calculated by using 60/(0.035 μm²). We note such high device density only represents the upper limit of our current fabrication process and cannot be realized in wafer size, because we ignore the larger trench area and vertical substrate area that are necessary for device fabrication. In the meantime, this theoretical integration density could be further increased by reducing the pitch size of the lateral device (using higher lithography resolution) or increasing the sidewall height (using deeper etching), as shown in Supplementary Fig. 7. On the other hand, in the second method (method #2), the device area is calculated by the total trench area, as schematically shown in Fig. 3c. Using this method, 60 vertical devices are located in a larger trench with area of 25 μm² (50 μm × 0.5 μm), leading to a device density of $2.4 \times 10^8$ cm$^{-2}$.

Furthermore, the high-density vertical devices could be scalable and realized by integrating devices onto multiple vertical sidewalls. To achieve this, we have fabricated device arrays with a pitch size of 420 μm in the x-direction and 90 μm in the y-direction (distance between two adjacent arrays), as shown in Fig. 3d, e. We note the array pitch size could be further reduced once the transistors are interconnected (without a large area of measurement pads). Next, the T-shape PDMS array is also fabricated with the same spacing, as shown in Fig. 3f. Finally, all transistors could be vertically laminated into the different trenches simultaneously with the assistance of the T-shape PDMS array (Fig. 3g), highlighting the potential of scalable vertical lamination.

### Layer-by-layer T-shape vertical integration

Due to the exact same structure between the T-shape PDMS stamp and the silicon sidewall, the vertical lamination process is highly robust and reproducible and could be repeated several times to realize layer-by-layer vertical lamination without consuming additional space or fabricating additional sidewalls. As shown in Fig. 4a, b, the first layer $MoS_2$ transistor array is vertically integrated onto the sidewall using the method described above. Next, atomic layer deposition is conducted to uniformly deposit the vertical interlayer dielectric. Afterwards, the second layer $MoS_2$ transistor array is further integrated onto the same sidewall using a T-shape lamination process, as illustrated in Fig. 4c.

This layer-by-layer vertical lamination could be further repeated for realizing high-density devices over a larger area. As a proof-of-concept demonstration, we have laminated 10 device layers successfully on the same vertical sidewall, as shown in Fig. 4d, e and Supplementary Fig. 8. With the assistance of layer-by-layer lamination, the device density is correspondingly increased using method #2, but would still remain the same if using method #1 for calculation. Since the trench structure could be batch-fabricated (as demonstrated in Fig. 3d), the realization of such device density over a large scale would be an interesting topic for further investigation.

Electrical characterizations are further conducted to examine the device qualities and the impact of layer-by-layer integration. The transfer curves and output curves of the $MoS_2$ transistors of three different layers are shown in Fig. 4f–h, and the corresponding device images are shown in Supplementary Fig. 9a–d. The devices in different vertical layers exhibit similar device performances (e.g., threshold voltage, on-state current), indicating our vertical lamination process won't alter the intrinsic properties of the bottom layer. Furthermore, to avoid electrical coupling, PMMA interlayer dielectric (~1 μm thick) is also laminated between each layer. As shown in Supplementary Fig. 9e–g, the device in the second layer exhibits consistent electrical performances with different gate voltages from the first layer, and its threshold voltage remains stable under various back gate voltages, indicating the low-κ PMMA layer could effectively reduce the interlayer electrical coupling.

## Discussion

In summary, we demonstrate a simple method to fabricate high-density vertical sidewall transistors by transferring pre-fabricated lateral transistors onto vertical substrates. Utilizing a custom-designed T-shape PDMS stamp lamination, the pre-fabricated transistors could conformally contact to the vertical substrate without damage or degradation, as verified by SEM, STEM, and electrical characterizations, hence overcoming the incompatibility between planar processes and vertical structures. With this technique, we have achieved a theoretical transistor density of up to $1.7 \times 10^{11}$ $cm^{-2}$ in the limited area of $0.035$ $μm^2$. Furthermore, we also provide two feasible approaches for the scalable fabrication of vertical sidewall transistor arrays, including simultaneous lamination onto multiple vertical substrates, as well as layer-by-layer laminations on the same vertical substrate. Our study realizes the stacking of multiple vertical transistors in a simple approach, opening up a new dimension for high-density integrated circuits.

## Methods

### Vertical substrates fabrication

Cr/Au (20/40 nm) patterns (width 500 μm and spacing 50 μm) are first fabricated on a Si (110) wafer through standard photolithography and high vacuum electron-beam evaporation. Next, the silicon is etched (using the Cr/Au layer as a mask) within 30% KOH solution at 100 °C for a few minutes to create deep vertical Si trenches. After the Cr/Au mask is removed by the $I_2$/KI and chromium solution, thermal oxidation is performed in a tube furnace under $O_2$ flow (150 sccm) at 1000 °C to obtain $SiO_2$ with a desired thickness, which is further measured and confirmed by focused ion beam and SEM characterization. We note that as the trench width decreases to below 3 μm, the T-shape PDMS stamp will be difficult to fit into the trench due to its poor visibility (high transparency) and low alignment accuracy (~1 μm), limiting the yield of the vertical lamination process.

### T-shape PDMS stamp preparation

A Si trench is treated by hexamethyldisilazane (HMDS) first to facilitate the subsequent release of the PDMS stamp from the Si mold (width 50 μm, as described in previous section). Next, PDMS elastomer (Sylgard 184, Dow Corning) is casted onto the Si mold to a thickness of 2 mm,

followed by baking at 60 °C for 12 hours and then peeling off from the mold to create the T-shape PDMS stamp with the same structure as the Si mold.

### Dry integration progress for sidewall device

First, a few-layer $MoS_2$ is mechanically exfoliated onto $SiO_2$ (300 nm) substrate, PMMA is spin-coated onto the substrate, and electrodes are defined via standard electron-beam lithography, followed by the deposition of 50 nm Au as contact electrodes (Supplementary Fig. 1a). Since the electrodes pass through vertical corners during vertical lamination process, it is necessary to choose the metal with good ductility, such as Au, Ag, Pt. Then, the wafer is functionalized by an HMDS layer in a sealed chamber at 80 °C for 10 mins and covered by a PMMA layer (~1 μm thick, Supplementary Fig. 1b). A PMMA gap is created by electron-beam lithography and development processes to release stress during the vertical lamination process (Supplementary Fig. 1c). The device layer can be picked up from the substrate by a flat PDMS stamp, transferred and released on top of the trench (Supplementary Fig. 1d–f). Subsequently, another T-shape PDMS stamp is pushed and fully embedded into the trench, thus the devices are pushed onto the vertical sidewall to form intimate contact between the device layer and the vertical substate (Supplementary Fig. 1g–i). For SEM characterizations, chloroform can be used to remove the PMMA layer (Supplementary Fig. 1j). To further reduce the wafer surface usage, we could laminate more devices with total length longer than the trench width. As shown in Supplementary Fig. 10, a 40 μm wide device array can still be pushed through a 27 μm long window.

### Materials characterizations and electrical measurements

SEM images are collected by TESCAN MIRA3 and VEGA3 at 20 kV, and cross-sectional STEM characterizations are performed using Thermo Scientific Themis Z 3.2, operating at 300 kV. To obtain better performance, the devices are annealed at 200 °C for two hours under an argon atmosphere[29]. Afterwards, electrical measurements are carried out in a probe station (Lakeshore, PS-100) at room temperature in a vacuum using an Agilent B1500A semiconductor analyzer. Note that the Si substrate is disconnected without applying voltage in the multilayer measurement. In addition, because the Si substrate is covered with a thick dielectric layer, electrical performance for each layer is largely unaffected by the Si substrate.

## Data availability

Relevant data supporting the key findings of this study are available within the article and the Supplementary Information file. All raw data generated during the current study are available from the corresponding authors upon request.

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

## Acknowledgements
Y.L. acknowledges the financial support from the National Key R&D Program of China (Grant No. 2021YFA1200503) and from the National Natural Science Foundation of China (Grant Nos. 51991340, 51991341, 61874041). Q.T. acknowledges the financial support from the China Postdoctoral Science Foundation (Grant No. 2023M741115). The authors acknowledge the Analytical Instrumentation Center of Hunan University for device characterization.

## Author contributions
Y.L. conceived the research. Y.L. Q.T. and R.W. designed the experiments. Q.T. led the device fabrication and electrical characterization. R.W., Wei L., and X.D. contributed to the CVD grown of MoS$_2$, SEM, and STEM characterizations. X.Z., Y.C., Wanying L., Z.L., L.M., X.Y., and Lei L. assisted in the device fabrication and electrical measurement. L.K., D.L., W.S., Liting L., S.D., and X.L. contributed to discussions and data analysis. Y.L. and Q.T. co-wrote the paper. All authors discussed the results and commented on the manuscript.

## Competing interests
The authors declare no competing interests.
