## [Peer Review File · Nature Communications]

High-density vertical sidewall MoS₂ transistors through T-shape vertical laminationREVIEWER COMMENTS

Reviewer #1 (Remarks to the Author):

The presented paper is extremely interesting for the demonstration of TMDC transfer to vertical sidewalls and it is to my feeling indeed first-time reporting of it. These results are relevant for publication. However it will need considerable corrections.

The English in the paper is readable but certain sections are difficult to understand. Most errors seem to be in the adverb choice and smaller grammar mistakes. Before being accepted it needs a serious review of the language.

However before doing so a more important modification seems needed in my opinion. As mentioned, the paper is extremely interesting from the point of view of TMDC transfer on vertical sidewalls. However, the main argument for applying the sidewall transfer is the area scaling, and this one does not stand in all cases due to some too optimistic assumptions.

To allow the suggested transfer technique, you still need to have a rectangular gap with minimal width of the sidewall height, limiting the wafer surface usage. This loss of wafer surface is not considered for the area scaling benefit calculation in the first 2 sections of the paper (signal and multiple sidewall transfer) and should be accounted for correctly. A fully correct calculation would also require accounting for the gap in the PMMA needed for the transfer and a contribution for the horizontal part needed for the contacts to the higher metal levels needed in the final technology. Strictly speaking the access lines to the devices that take space in the other direction (Y) would also have to be accounted for, but if the circuits are transferred this will be a small contribution. I think there is a real area scaling benefit in the case of the multiple stacked layer transfer that was also demonstrated, but also there the area scaling benefit has to be assessed correctly considering the lost horizontal area on the receiving wafer.

The demonstrated transfer cases and the electrical data are well chosen to my feeling to demonstrate the potential, yield an integrity of the transfer method.

Some more specific feedback is below.

The section on p3 might be a bit too restrictive and seems limited to lift-off based device integration. To my feeling there is nothing intrinsically unscalable in a correctly integrated stacked device like CFET, but it is complex indeed. The authors claim that vertically stacked devices are a demonstration only architecture but that is not true. Si are currently being developed, with many papers published by major IEDM on the topic. It is also considered for 2D (see also: <https://doi.org/10.1038/s41467-023-41779-5> and reference [22]). These architectures might have the same or even better area scaling as the suggested ones and can hence not be ignored.

In figure 1 it would be good to add some info on the PMMA gap creation (there is some in extra material) but it would be better to add it to the legend if figure 1.

I think Figure 4 needs some revision, in view of the comment above. The width of the gap has to be enough to allow also the last transfer it is not the case in the current figure. (eventually a one sidewall image would do the job)

There were some extra questions that could require some development for improved relevance.

- The transfer over the corner requires the contact and gate metals to be ductile. Au and Bi seem to be ok for that. But it might be worth a comment.

- Can the authors comment on the expected scaling of the trench width and density to which the PDMS stamp creation could work. This could help the reader to assess the potential target technology.

Reviewer #2 (Remarks to the Author):

The manuscript entitled " High-density vertical sidewall MoS₂ transistors through T-shape vertical lamination" by Tao et al. report an approach to fabricate FET vertically by PDMS stamping method. Although the approach is quite interesting, the main claim of the study is not so convincing. The density of devices was calculated by the vertical size of the devices over the lateral area, which make the density value is enormous. Nevertheless, the lateral devices can be fabricated with multilayer, giving rise to the same value of bulk density to the proposing approach. The device performance of the study is much higher than 66mV/dec and the on-current density is not so impressive (as I estimate). Therefore, the study should be suitable for other journals rather than Nature Communication.

In addition, I have other concerns.

1. Normally, each FETs requires its own gate. The array of devices demonstrated in this study uses the same gate. In Figure 3, is Cr/Au a common gate for all devices? If so, is it possible to fabricate the separated gates for each device?
2. What is the performance of the device in Figure 3c? With the high-k gate dielectric HfO₂, I guess the performance of the device should be better than the normal SiO₂.
3. In Figure 4f, do the authors use the bottom extra Au electrodes of the first layer as a gate for second electrodes? If so, the applied gate bias will affect first layer devices. Why do the authors not make separated gates for each layer and then PDMS laminating them?

Reviewer #3 (Remarks to the Author):

Quanyang Tao et al. reported vertical laminated MoS₂ sidewall transistors with high device density. The authors have successfully demonstrated a high feasibility for scalable fabrication approach not only for multiple stacking but also large-scaled device production. Although the research primarily focused on the device fabrication approach, the presented results are quite interesting for a future 2D-based device fabrication methodology and are reserved to be further considered in Nature Communications after following discussions.

Q1. The authors claimed that the vertically fabricated transistors remain free from damage or degradation after lamination using T-shape PDMS, and they also reported low device-to-device variation. However, there is a slight degradation in carrier mobility and on-off current ratio, as observed in Figure 2c. To address this issue more conclusively, a TEM surface analysis should be conducted. Additionally, it is recommended that further statistical electrical performance analysis be provided, based on the simultaneous fabrication of numerous devices, to substantiate the authors' claim.

Q2. What is the maximum depth of a silicon trench that ensures a surface free of any damage concerning electrical device performance? This information is crucial for estimating the maximum device density and understanding the limitations of this approach. Please provide further details on the maximum device density including estimation procedures. Additionally, it would be helpful to discuss specifically how the device density was calculated for both lateral and vertical MoS₂ devices in the manuscript.

Q3. The transconductance curves corresponding to Fig. 2a and Supplementary Fig. 3 should be provided to gain a detailed understanding of the carrier transport mechanism and to estimate field-effect mobility as a function of drain voltage. In particular, what is the origin of first and second humps in transfer curve of supplementary Fig. 3? Is this relating to a vertical conducting channel migration? Please discuss.

Q4. How the authors confirm the thickness of sidewall gate oxide?

Q5. Please analytically define the voltage gain.

Q6. The main reason for the positive shift in turn-on voltage after layer-by-layer laminations, as observed in Figures 4g-h, should be thoroughly discussed. Given the authors' emphasis on preserving the intrinsic properties of MoS₂ after lamination, it is essential to explore this observed voltage shift. This discussion should consider factors such as enhanced subthreshold swing or suppressed interface trap density, in addition to gate coupling effects. Providing stacking (lamination) number-dependent transfer curves would be beneficial for readers to gain a comprehensive understanding.

Responses to Reviewer #1:

General comments: The presented paper is extremely interesting for the demonstration of TMDC transfer to vertical sidewalls and it is to my feeling indeed first-time reporting of it. These results are relevant for publication. However it will need considerable corrections.

The English in the paper is readable but certain sections are difficult to understand. Most errors seem to be in the adverb choice and smaller grammar mistakes. Before being accepted it needs a serious review of the language.

However before doing so a more important modification seems needed in my opinion. As mentioned, the paper is extremely interesting from the point of view of TMDC transfer on vertical sidewalls. However, the main argument for applying the sidewall transfer is the area scaling, and this one does not stand in all cases due to some too optimistic assumptions.

To allow the suggested transfer technique, you still need to have a rectangular gap with minimal width of the sidewall height, limiting the wafer surface usage. This loss of wafer surface is not considered for the area scaling benefit calculation in the first 2 sections of the paper (signal and multiple sidewall transfer) and should be accounted for correctly. A fully correct calculation would also require accounting for the gap in the PMMA needed for the transfer and a contribution for the horizontal part needed for the contacts to the higher metal levels needed in the final technology. Strictly speaking the access lines to the devices that take space in the other direction (Y) would also have to be accounted for, but if the circuits are transferred this will be a small contribution.

I think there is a real area scaling benefit in the case of the multiple stacked layer transfer that was also demonstrated, but also there the area scaling benefit has to be assessed correctly considering the lost horizontal area on the receiving wafer.

The demonstrated transfer cases and the electrical data are well chosen to my feeling to demonstrate the potential, yield an integrity of the transfer method. Some more specific feedback is below.

Response: We thank reviewer for the positive comments “the presented paper is extremely interesting”, and “it is to my feeling indeed first-time reporting of it”. We particularly appreciate the highly insightful and constructive comments from the reviewer and welcome the opportunity to address these questions and describe the revisions we have made accordingly, as below.

First of all, we fully agree with the reviewer that we “need to have a rectangular gap with minimal width of the sidewall height, limiting the wafer surface usage”. This is because in our previous method, the maximum device length (that can be laminated onto the vertical sidewall) is determined by the width of the trench, as shown in Fig. R1a. To overcome this limitation, we have now extended our method by using devices longer than the trench width, as schematically shown in Fig. R1b. Within our improved method, the trench depth (40 μm) is larger than the trench width (27 μm). By using vertical T-shape PDMS, 40 μm wide device array can still be pushed through 27 μm long window. This proof-of-concept experiment demonstrates the potential for further area scaling of the sidewall devices.

Furthermore, we also highly appreciate the reviewer’s suggestion that the lost lateral area should also be considered during the density calculation. Taking this suggestion, we have re-calculated the real device density using this method, and clearly

stated the difference between theoretical density (in small trench area) and the real density (with lateral area accounted), as detailed in the revision below. In the meantime, we have also followed the reviewer's suggestion by fixing the gramma mistake and typos throughout our manuscript.

Fig. R1. Vertical transfer processes of devices. **a**, Previous transfer method, the lateral distance of the devices is determined by the trench width. **b**, Improved transfer method by using devices longer than the trench width. Scale bars are 10 μm.

Revision:

1. In page 9, line 198 of main manuscript, we included the following discussion: “The device density could be further calculated using device number divided by total device area. Here, the device number is 60, and the device area is defined using two different methods. In the first method (method #1), the total device area is the planar region of one device, which is $\sim 0.035 \mu\text{m}^2$ ($70.7 \text{ nm} \times 0.5 \mu\text{m}$) including all active components (gate electrode, dielectric layer, MoS_2 channel and source/drain electrodes), as schematically illustrated in Fig. 3c. Therefore, a highest density of $1.7 \times 10^{11} \text{ cm}^{-2}$ could be calculated by using $60 / (0.035 \mu\text{m}^2)$. We note such high device density only represents the upper limit of our current fabrication process and cannot be realized in wafer size, because we ignore the larger trench area and vertical substrate area that are necessary for device fabrication. In the meantime, this theoretical integration density could be further increased by reducing the pitch size of the lateral device (using higher lithography resolution) or increasing the sidewall height (using deeper etching), as shown in Supplementary Fig. 7. On the other hand, in the second method (method #2), the device area is calculated by the total trench area, as schematically shown in Fig. 3c. Using this method, 60 vertical devices are located in a larger trench with area of $25 \mu\text{m}^2$ ($50 \mu\text{m} \times 0.5 \mu\text{m}$), leading to a device density of $2.4 \times 10^8 \text{ cm}^{-2}$.”

2. In page 11, line 224 of main manuscript, we revised Fig. 3.

3. In page 15, line 313 of main manuscript, we added the following discussion: “To further reduce the wafer surface usage, we could laminate more devices with total length longer than the trench width. As shown in Supplementary Fig. 10, 40 μm wide device array can still be pushed through 27 μm long window.”

4. In page 30, line 471 of Supplementary information, we added Supplementary Fig. 10.

Specific Comment 1. The section on p3 might be a bit too restrictive and seems limited to lift-off based device integration. To my feeling there is nothing intrinsically unscalable in a correctly integrated stacked device like CFET, but it is complex indeed. The authors claim that vertically stacked devices are a demonstration only architecture but that is not true. Si are currently being developed, with many papers published by major IEDM on the topic. It is also considered for 2D (see also: <https://doi.org/10.1038/s41467-023-41779-5> and reference [22]). These architectures might have the same or even better area scaling as the suggested ones and can hence not be ignored.

Response: We thank the reviewer for this insightful question. We fully agree with the reviewer that stacked devices are being developed using Si and 2D channels, as pointed out by the references [*IEEE IEDM*, 2020, 20.26.21 (2020); *IEEE IEDM* 7.5.1 (2022); *Nat. Commun.* 14, 6400 (2023)]. However, stacking high-density devices (in vertical direction) is still very challenging because the fabrication process needs to be repeated for each device layer. This is indeed the primary motivation of our study here, to develop an alternative route for constructing multiple devices in vertical direction simultaneously. Within our proof-of-concept process, multiple lateral transistors could be pre-fabricated on planar substrate first, and then dry laminated onto the vertical substrate through a custom-designed T-shape stamp, hence avoiding the repetitive processes for multiple stacked devices.

We thank the reviewer for this question, and we have further clarified the motivation of our process in the revised introduction part.

Specific Comment 2. In figure 1 it would be good to add some info on the PMMA gap creation (there is some in extra material) but it would be better to add it to the legend in figure 1.

Response: We thank the reviewer for raising up this insightful suggestion. We have now schematically illustrated the PMMA gap creation process. As shown in Fig. R2a, after spin-coating PMMA layer, the desired PMMA gap can be created through electron-beam lithography and development processes. Following the reviewer’s suggestion, we have added an optical image with the PMMA gap and described it in the legend in Fig. 1 (as shown in Fig. R2b, iii below).

Fig. R2. The PMMA gap creation and vertical lamination processes. **a**, Schematics of the PMMA gap creation, including PMMA sin-coating, electron-beam lithography and development processes. **b**, Schematics and the corresponding optical images of the vertical lamination processes with 5 steps: planar MoS₂ transistors pre-fabricated on a sacrificial substrate (i), fabrication of vertical silicon trench through etching (ii), dry-transfer of MoS₂ transistors with the PMMA gap on top of Si trench (iii), T-shape PDMS stamp laminated and pushed into the trench (iv), and MoS₂ vertical transistors after lamination (vi). Scale bars are 10 μm for panel i, iii, vi, and 20 μm for panel ii, iv.

Revision:

1. In page 6, line 120 of main manuscript, we revised Fig. 1.
2. In page 15, line 305 of the revised Method section, we added the following description: “A PMMA gap is created by electron-beam lithography and development processes to release stress during the vertical lamination process (Supplementary Fig. 1c).”
3. In page 21, line 418 of Supplementary information, we added the schematics of the PMMA gap creation in Supplementary Fig. 1.”

Specific Comment 3. I think Figure 4 needs some revision, in view of the comment above. The width of the gap has to be enough to allow also the last transfer it is not the case in the current figure. (eventually a one sidewall image would do the job)

Response: We thank the reviewer for this question, and we have now realized our schematics in Fig. 4 could lead to confusion for readers. Based on the reviewer’s suggestion, we have now revised the corresponding schematics using only one sidewall, as shown in Fig. R3 below.

Fig. R3. Layer-by-layer vertical integrations of high-density vertical devices. **a-d**, Schematics of the 10 times layer-by-layer vertical laminations of transistors on the same sidewall.

Revision:

In page 13, line 257 of main manuscript, we revised the Fig. 4.

Specific Comment 4. There were some extra questions that could require some development for improved relevance.

- The transfer over the corner requires the contact and gate metals to be ductile. Au and Bi seem to be ok for that. But it might be worth a comment.

- Can the authors comment on the expected scaling of the trench width and density to which the PDMS stamp creation could work. This could help the reader to assess the potential target technology.

Response: We thank the reviewer for pointing out these important questions. First of all, we fully agree with the reviewer that “The transfer over the corner requires the contact and gate metals to be ductile”, and we have now clearly mentioned this point in the revised manuscript.

Furthermore, the smallest trench width is $\sim 3 \mu\text{m}$ in our current approach. With further reducing the trench width below $3 \mu\text{m}$, the T-shape PDMS stamp will be difficult to fit into the trench due to its poor visibility (high transparency) and low alignment accuracy ($\sim 1 \mu\text{m}$), limiting the yield of the vertical lamination process.

We thank the reviewer for these questions, and we have further discussed the metal electrode ductility and the scaling of trench width in the revised manuscript.

Revision:

1. In page 15, line 301 of main manuscript, we added the following discussion: “Since the electrodes pass through vertical corners during vertical lamination process, it is necessary to choose the metal with good ductility, such as Au, Ag, Pt.”

2. In page 14, line 288 of main manuscript, we added the following discussion: “We note that as the trench width decreases to below $3 \mu\text{m}$, the T-shape PDMS stamp will be difficult to fit into the trench due to its poor visibility (high transparency) and low alignment accuracy ($\sim 1 \mu\text{m}$), limiting the yield of the vertical lamination process.”

Responses to Reviewer #2:

General comments: The manuscript entitled "High-density vertical sidewall MoS₂ transistors through T-shape vertical lamination" by Tao et al. report an approach to fabricate FET vertically by PDMS stamping method. Although the approach is quite interesting, the main claim of the study is not so convincing. The density of devices was calculated by the vertical size of the devices over the lateral area, which make the density value is enormous. Nevertheless, the lateral devices can be fabricated with multilayer, giving rise to the same value of bulk density to the proposing approach. The device performance of the study is much higher than 66mV/dec and the on-current density is not so impressive (as I estimate). Therefore, the study should be suitable for other journals rather than Nature Communication. In addition, I have other concerns.

Response: We thank reviewer for the recognition that “the approach is quite interesting”. We particularly appreciate the highly insightful and constructive comments from the reviewer and welcome the opportunity to address these questions and describe the revisions we have made accordingly, as below.

First of all, regarding to device density, we agree with the reviewer that “the density of devices was calculated by the vertical size of the devices over the lateral area”. In fact, our claimed current density only represents the theoretical highest density within a very limited area ($0.035 \mu\text{m}^2$), while the real device density could be much lower by counting the lateral trench size. We have now realized our calculation method could lead to confusion for readers. To clarify this question, we have now thoroughly revised our manuscript and clarified the real device density, as detailed in the revision below.

Furthermore, besides the device density, the major advantage of our approach is to integrate multiple stacked devices in a simple way. We agree with the reviewer that “the lateral devices can be fabricated with multilayer”. However, vertically stacking multilayer lateral devices (in vertical direction) is still very challenging because the fabrication process needs to be repeated for each device layer. This is indeed the primary motivation of our study here, to develop an alternative route for constructing multiple devices in vertical direction. Within our proof-of-concept process, multiple lateral transistors could be pre-fabricated on planar substrate first, and then dry laminated onto the vertical substrate through a custom-designed T-shape stamp, hence avoiding the repetitive processes for multiple stacked devices.

Finally, regarding to the device performance, our previous device is not optimized (due to long channel) because the primary motivation of our work is to demonstrate a new way for fabricating multiple devices in vertical direction. Based on the reviewer’s question, we have now fabricated additional device with shorter channel length of 150 nm and thinner gate dielectric of 10 nm Al_2O_3 . As shown in Fig. R4, the vertical MoS_2 transistor exhibits an on-state current of up to $400 \mu\text{A}/\mu\text{m}$, small SS of 110 mV/dec and high on-off ratio of 10^9 , comparable with the state-of-the-art MoS_2 planar transistors [*Nature* 593, 211 (2021); *IEEE IEDM* 37.3.1 (2021)].

Fig. R4. Electrical characteristics of vertical MoS_2 transistor with 150 nm channel length using 10 nm thick Al_2O_3 as side gate dielectric. a, b, I_{ds} – V_{gs} transfer characteristics (a) and I_{ds} – V_{ds} output curves (b) of the vertical MoS_2 transistor using stronger gate control.

Revision:

1. In page 9, line 198 of main manuscript, we included the following discussion: “The device density could be further calculated using device number divided by total device area. Here, the device number is 60, and the device area is defined using two different methods. In the first method (method #1), the total device area is the planar region of one device, which is $\sim 0.035 \mu\text{m}^2$ ($70.7 \text{ nm} \times 0.5 \mu\text{m}$) including all active components

(gate electrode, dielectric layer, MoS₂ channel and source/drain electrodes), as schematically illustrated in Fig. 3c. Therefore, a highest density of $1.7 \times 10^{11} \text{ cm}^{-2}$ could be calculated by using $60 / (0.035 \text{ } \mu\text{m}^2)$. We note such high device density only represents the upper limit of our current fabrication process and cannot be realized in wafer size, because we ignore the larger trench area and vertical substrate area that are necessary for device fabrication. In the meantime, this theoretical integration density could be further increased by reducing the pitch size of the lateral device (using higher lithography resolution) or increasing the sidewall height (using deeper etching), as shown in Supplementary Fig. 7. On the other hand, in the second method (method #2), the device area is calculated by the total trench area, as schematically shown in Fig. 3c. Using this method, 60 vertical devices are located in a larger trench with area of $25 \text{ } \mu\text{m}^2$ ($50 \text{ } \mu\text{m} \times 0.5 \text{ } \mu\text{m}$), leading to a device density of $2.4 \times 10^8 \text{ cm}^{-2}$.”

2. In page 11, line 224 of main manuscript, we revised Fig. 3.

3. In page 9, line 195 of main manuscript, we added the following sentence: “The electrical measurements of short-channel vertical devices are shown in Supplementary Fig. 6, exhibiting on-state current of up to $400 \text{ } \mu\text{A}/\mu\text{m}$, small SS of $110 \text{ mV}/\text{dec}$ and high on-off ratio of 10^9 .”

4. In page 26, line 452 of Supplementary information, we added Supplementary Fig. 6.

Specific Comment 1. Normally, each FETs requires its own gate. The array of devices demonstrated in this study uses the same gate. In Figure 3, is Cr/Au a common gate for all devices? If so, is it possible to fabricate the separated gates for each device?

Response: We thank the reviewer for raising up these questions. In our previous experiment, we use global side gate, where all devices share a common gate electrode. Following the reviewer’s suggestion, we have fabricated separated gates for MoS₂ devices, using $10 \text{ nm Al}_2\text{O}_3$ as the interlayer dielectric, as shown in Fig. R5a, b. The vertical MoS₂ transistor with the separated gate exhibits decent n-type behaviors with on-off ratio over 10^8 , as well as linear $I_{\text{ds}}-V_{\text{ds}}$ output characteristics (Fig. R5c, d).

We thank the reviewer for these questions, and we have included the vertical MoS₂ transistors with separated gates in the revised manuscript.

Fig. R5. Electrical characteristics of vertical MoS₂ transistors with separated gates. **a**, SEM image of separated gates on the vertical sidewall. **b**, SEM image of vertical MoS₂ transistors with separated gates. Inset is the cross-sectional schematic of the vertical MoS₂ transistors. **c**, **d**, $I_{\text{ds}}-V_{\text{gs}}$ transfer characteristics (**c**) and $I_{\text{ds}}-V_{\text{ds}}$ output curves (**d**) of the vertical MoS₂ transistor with the separated gate.

Revision:

1. In page 7, line 156 of main manuscript, we included the following discussion: “Besides global side-gate devices, we could also fabricate separated gates for vertical MoS₂ devices using $10 \text{ nm Al}_2\text{O}_3$ as the dielectric, as shown in Supplementary Fig. 4.”

2. In page 24, line 443 of Supplementary information, we added Supplementary Fig. 4.

Specific Comment 2. What is the performance of the device in Figure 3c? With the high-k gate dielectric HfO₂, I guess the performance of the device should be better than the normal SiO₂.

Response: We thank the reviewer for this question. We have now included electrical performance of devices in Fig. 3c (as shown in Fig. R6 below). The reviewer is corrected, with high-k gate dielectric (10 nm thick Al₂O₃), the vertical MoS₂ transistor exhibits higher electrical performance than the planar transistor on normal SiO₂ (300 nm thick), including smaller SS, larger on-state current, and weaker DIBL effect.

We thank the reviewer for this question, and we have now included the performance of the short-channel length device (of Fig. 3c) in the revised manuscript.

Fig. R6. Electrical characterizations of MoS₂ transistors with 150 nm channel length. a, b, I_{ds} – V_{gs} transfer characteristics (a) and I_{ds} – V_{ds} output curves (b) of the planar MoS₂ transistor with 150 nm channel length on 300 nm SiO₂. **c, d,** I_{ds} – V_{gs} transfer characteristics (c) and I_{ds} – V_{ds} output curves (d) of the vertical MoS₂ transistor with 150 nm channel length on 10 nm Al₂O₃.

Revision:

1. In page 9, line 195 of main manuscript, we added the following sentence: “The electrical measurements of short-channel vertical devices are shown in Supplementary Fig. 6, exhibiting on-state current of up to 400 μ A/ μ m, small SS of 110 mV/dec and high on-off ratio of 10^9 .”

2. In page 26, line 452 of Supplementary information, we added Supplementary Fig. 6.

Specific Comment 3. In Figure 4f, do the authors use the bottom extra Au electrodes of the first layer as a gate for second electrodes? If so, the applied gate bias will affect first layer devices. Why do the authors not make separated gates for each layer and then PDMS laminating them?

Response: We thank the reviewer for these excellent questions. In previous Fig. 4f, the first layer and second layer device share a common electrode (that is, the first layer

source-drain is used as the second layer gate) to reduce the vertical device area (thickness in planar direction). Following the reviewer’s suggestion, we have now fabricated three-layer MoS₂ transistors with separated gates through layer-by-layer integration, to avoid interactions between devices in different layers, as shown in Fig. R7a–d. Each layer device exhibits consistent electrical performance (Fig. R7e–g), indicating our vdW vertical lamination process won’t alter the intrinsic properties of bottom layer.

We thank the reviewer for these questions, and we have included the devices without sharing common electrodes in the revised manuscript.

Fig. R7. Layer by layer integrated MoS₂ transistors with separated gates using BN flakes as the dielectrics. **a**, Optical image of the suspended first-layer MoS₂ transistor. **b**, Optical image of the suspended two-layer MoS₂ transistors. **c**, Optical image of the suspended three-layer MoS₂ transistors. The red and white dotted boxes are MoS₂ and BN in the first-layer (**a**), second-layer (**b**) and third-layer (**c**) device, respectively. **d**, Optical image of the vertical three-layer MoS₂ transistors. Inset is the cross-sectional schematic of the vertical three-layer devices. Scale bars are 10 μm. **e–f**, I_{ds} – V_{gs} transfer curves and I_{ds} – V_{ds} output curves (inset) of the first-layer (**e**), second-layer (**f**) and third-layer (**g**) MoS₂ transistor.

Revision:

1. In page 12, line 251 of main manuscript, we revised the following sentence: “The transfer curves and output curves of the MoS₂ transistors of three different layers are shown in Fig. 4f–h, and the corresponding device images are shown in Supplementary Fig. 9. The devices in different vertical layers exhibit similar device performances (e.g., threshold voltage, on-state current), indicating our vdW vertical lamination process won’t alter the intrinsic properties of bottom layer.”

2. In page 29, line 463 of Supplementary information, we added Supplementary Fig. 9.

Responses to Reviewer #3:

General comments: Quanyang Tao et al. reported vertical laminated MoS₂ sidewall transistors with high device density. The authors have successfully demonstrated a high feasibility for scalable fabrication approach not only for multiple stacking but also large-scaled device production. Although the research primarily focused on the device fabrication approach, the presented results are quite interesting for a future 2D-based device fabrication methodology and are reserved to be further considered in Nature Communications after following discussions.

Response: We thank reviewer for the positive comments “have successfully demonstrated a high feasibility for scalable fabrication approach” and “the presented results are quite interesting”. We also appreciate the specific questions raised and would like to take this opportunity to further clarify these questions as below.

Specific Comment 1. The authors claimed that the vertically fabricated transistors remain free from damage or degradation after lamination using T-shape PDMS, and they also reported low device-to-device variation. However, there is a slight degradation in carrier mobility and on-off current ratio, as observed in Figure 2c. To address this issue more conclusively, a TEM surface analysis should be conducted. Additionally, it is recommended that further statistical electrical performance analysis be provided, based on the simultaneous fabrication of numerous devices, to substantiate the authors' claim.

Response: We thank the reviewer for these insightful suggestions. Following the reviewer's suggestion, we have fabricated 60 lateral MoS₂ transistors and 60 vertical MoS₂ transistors, and statistically extracted the on-state current and field effect mobility of the devices. As shown in Fig. R8a–d, the key electrical properties remain consistent between lateral devices and vertical devices, further confirming the transistors remain free from damage or degradation after vertical lamination. In addition, we have also included the TEM characterization of the vertical devices (Fig. R8e), demonstrating the device layer in conformal contact with the vertical substrate without any visible damage.

We thank the reviewer for these questions, and we have added statistical electrical performance analysis of MoS₂ devices (on lateral and vertical substrates) in the revised manuscript.

Fig. R8. Statistical analysis of electrical performance of MoS₂ transistors and structural characterization. **a, b**, $I_{ds}-V_{gs}$ transfer characteristics of 60 lateral MoS₂ devices (**a**) and 60 vertical MoS₂ devices (**b**) at $V_{ds}=1$ V. **c, d**, Statistical distribution of on-state current and field effect mobility of MoS₂ devices in **a** and **b**, respectively. **e**, Cross-sectional TEM image of the vertical MoS₂ transistor.

Revision:

1. In page 7, line 152 of main manuscript, we added the following discussion: “To further investigate the device uniformity, we have fabricated 60 planar and vertical transistors, and statistically analyzed their transfer curves. As shown in Supplementary Fig. 3, their key electrical properties demonstrate low device-to-device variations, further confirming the transistors remain free from damage or degradation after vertical lamination.”

2. In page 23, line 437 of Supplementary information, we added Supplementary Fig. 3.

Specific Comment 2. What is the maximum depth of a silicon trench that ensures a surface free of any damage concerning electrical device performance? This information is crucial for estimating the maximum device density and understanding the limitations of this approach. Please provide further details on the maximum device density including estimation procedures. Additionally, it would be helpful to discuss specifically how the device density was calculated for both lateral and vertical MoS₂ devices in the manuscript.

Response: We thank the reviewer for these excellent questions. We fully agree with the reviewer that the calculation of the device density is important and “it would be helpful to discuss specifically how the device density was calculated”. Based on the reviewer’s suggestion, we have now thoroughly discussed the density calculation for both lateral devices and vertical devices, and included this discussion in the revised manuscript, as follows.

First of all, the density of lateral devices is calculated using device number divided by total device area. As schematically shown in Fig. R9a, the total device area includes the device channel area and the source-drain contact area, but does not include the large measurement pad (for electrical test). The device number is defined as the number of separate channels, where different channels could share the same contact electrodes (e.g., we have 4 devices in Fig. R9a).

Furthermore, two different methods are used to calculate the device density of vertical devices. Within these two methods, the device number is the same as lateral devices (before vertical lamination), and the difference is the definition of device area. In method #1, the device area is calculated by the lateral area of the pure devices, as schematically illustrated in Fig. R9b. With our approach, we could integrate 60 devices within a small area of $\sim 0.035 \mu\text{m}^2$ (length of 70 nm and width of $0.5 \mu\text{m}$ in Fig. R9b), leading to a high device density of $1.7 \times 10^{11} \text{cm}^{-2}$ using this method. We note high device density could only exist in a small area of $0.035 \mu\text{m}^2$ and can’t be realized in wafer size, because we ignore the larger trench area that is necessary for device fabrication. We also note by further layer-by-layer vertical lamination, this density won’t increase because the total device area and device number will increase at the same time.

In contrast, in method #2, the device area is calculated by the total trench area, as shown in Fig. R9b. Using this method, the total device density would be a lower number. For example, for a single layer vertical lamination, 60 vertical devices are located in a larger trench of $50 \mu\text{m}$, leading to a device density of $2.4 \times 10^8 \text{cm}^{-2}$. By further laminating 10 device layers in the trench (Fig. R9c), the density could increase by an

order of magnitude to $6 \times 10^9 \text{ cm}^{-2}$. Although device density using this method is much lower, it could be more practical since such device density could be realized in wafer scale. In addition, we also note the trench depth is $\sim 50 \mu\text{m}$ in our method. Increasing the trench depth will increase the total device number, but would also need larger trench width for vertical lamination. Hence, the trench depth won't impact the device density too much using method #2.

We thank the reviewer for these questions, and we have further provided detailed calculation methods for both lateral and vertical devices in the revised manuscript.

Fig. R9. Definition of device area. **a**, The area of lateral devices. **b**, Two definition methods for vertical device area. **c**, The area of 10-layer vertical devices, where each device layer contains 60 transistors.

Revision:

1. In page 9, line 198 of main manuscript, we included the following discussion: “The device density could be further calculated using device number divided by total device area. Here, the device number is 60, and the device area is defined using two different methods. In the first method (method #1), the total device area is the planar region of one device, which is $\sim 0.035 \mu\text{m}^2$ ($70.7 \text{ nm} \times 0.5 \mu\text{m}$) including all active components (gate electrode, dielectric layer, MoS_2 channel and source/drain electrodes), as schematically illustrated in Fig. 3c. Therefore, a highest density of $1.7 \times 10^{11} \text{ cm}^{-2}$ could be calculated by using $60/(0.035 \mu\text{m}^2)$. We note such high device density only represents the upper limit of our current fabrication process and cannot be realized in wafer size, because we ignore the larger trench area and vertical substrate area that are necessary for device fabrication. In the meantime, this theoretical integration density could be further increased by reducing the pitch size of the lateral device (using higher lithography resolution) or increasing the sidewall height (using deeper etching), as shown in Supplementary Fig. 7. On the other hand, in the second method (method #2), the device area is calculated by the total trench area, as schematically shown in Fig. 3c. Using this method, 60 vertical devices are located in a larger trench with area of $25 \mu\text{m}^2$ ($50 \mu\text{m} \times 0.5 \mu\text{m}$), leading to a device density of $2.4 \times 10^8 \text{ cm}^{-2}$.”

2. In page 11, line 224 of main manuscript, we revised Fig. 3.

Specific Comment 3. The transconductance curves corresponding to Fig. 2a and Supplementary Fig. 3 should be provided to gain a detailed understanding of the carrier transport mechanism and to estimate field-effect mobility as a function of drain voltage. In particular, what is the origin of first and second humps in transfer curve of supplementary Fig. 3? Is this relating to a vertical conducting channel migration? Please discuss.

Response: We thank the reviewer for these suggestions. The humps in the transfer curve could be attributed to the non-uniform doping within the metal- MoS_2 contact, leading to different contact barrier within the same device. To improve the contact and

to remove the humps in transfer curve, we have annealed the device at 200 °C for two hours. As shown in Fig. R10a, b, the humps in transfer curves clearly disappear after annealing. Furthermore, following the reviewer’s suggestion, we have provided the transconductance (G_m) curves of 60 MoS₂ devices on both lateral and vertical substrates after annealing (Fig. R10c, d), exhibiting consistent device performances.

Thanks for the reviewer’s these questions, and we have included the device data using annealing process and provided the transconductance curves of MoS₂ devices in the revised manuscript.

Fig. R10. Electrical characteristics of MoS₂ transistors. a, b, I_{ds} - V_{gs} transfer curves of the vertical MoS₂ transistors before (a) and after (b) annealing. c, d, The transconductance curves of lateral MoS₂ devices (c) and vertical MoS₂ devices (d).

Revision:

1. In page 23, line 437 of Supplementary information, we added Supplementary Fig. 3.
2. In page 15, line 320 of main manuscript, we added the following sentence: “To obtain better performance, the devices are annealed at 200 °C for two hours under argon atmosphere.”

Specific Comment 4. How the authors confirm the thickness of sidewall gate oxide?

Response: We thank the reviewer for raising up this important question. The sidewall gate oxide is confirmed using cross-sectional SEM image, as shown in Fig. R11. We thank the reviewer for this question, and we have further explained the sidewall gate oxide calculation in the revised manuscript.

Fig. R11. Cross-sectional SEM image of the vertical sidewall, where the thickness of sidewall gate oxide is ~300 nm.

Revision:

In page 14, line 286 of main manuscript, we revised the following sentence: “Thermal oxidation is performed in a tube furnace under O₂ flow (150 sccm) at 1000 °C to obtain SiO₂ with desired thickness, which is further measured and confirmed by focused ion beam and SEM characterization.”

Specific Comment 5. Please analytically define the voltage gain.

Response: We thank the reviewer for this suggestion, and we have defined the voltage gain ($\text{Gain} = d(V_{\text{out}})/d(V_{\text{in}})$) in the revised manuscript.

Revision:

In page 8, line 163 of main manuscript, we revised the following sentence: “The inverter demonstrates sharp output voltage transition as a function of input voltage, yielding a voltage gain (defined as $|dV_{\text{out}}/dV_{\text{in}}|$) of 10 at $V_{\text{dd}} = 2 \text{ V}$ (Fig. 2e).”

Specific Comment 6. The main reason for the positive shift in turn-on voltage after layer-by-layer laminations, as observed in Figures 4g-h, should be thoroughly discussed. Given the authors' emphasis on preserving the intrinsic properties of MoS₂ after lamination, it is essential to explore this observed voltage shift. This discussion should consider factors such as enhanced subthreshold swing or suppressed interface trap density, in addition to gate coupling effects. Providing stacking (lamination) number-dependent transfer curves would be beneficial for readers to gain a comprehensive understanding.

Response: We thank the reviewer for these insightful suggestions. First of all, in our previous Fig. 4g, h, the devices in first layer and second layer have different device structures, where the first layer device has a top-gate structure while the second layer device has a back-gate structure, as shown in Fig. R12a, b. Therefore, the devices exhibit different threshold voltages and subthreshold swings, because the back gate modulates both channel and contact regions while the top gate typically only modulates the channel area.

Furthermore, to avoid such structure difference, we have fabricated additional devices using consistent top-gate structure (Fig. R12c–f). In the meantime, based on the reviewer’s suggestion, we have also provided the stacking (lamination) number-

dependent transfer curves. As shown in Fig. R12g–i, the devices in three different vertical layers exhibit similar device performances (e.g., threshold voltage, on-state current), indicating our vdW vertical lamination process won't alter the intrinsic properties of bottom layer.

We thank the reviewer for these suggestions, and we have further updated the Fig. 4 g, h (now Fig. 4f–h), and included the electrical performances of three-layer devices in the revised manuscript.

Fig. R12. Layer by layer integrated MoS₂ transistors. **a, b**, Schematics of the first layer device with top-gate structure (**a**) and the second layer device with back-gate structure (**b**), respectively. **c**, Optical image of the suspended first-layer MoS₂ transistor. **d**, Optical image of the suspended two-layer MoS₂ transistors. **e**, Optical image of the suspended three-layer MoS₂ transistors, the red and white dotted boxes are MoS₂ and BN in the first-layer (**c**), second-layer (**d**) and third-layer (**e**) device, respectively. **f**, Optical image of the vertical three-layer MoS₂ transistors. Scale bars are 10 μm . Inset is the cross-sectional schematic of the vertical three-layer devices. **g–i**, $I_{\text{ds}}-V_{\text{gs}}$ transfer curves and $I_{\text{ds}}-V_{\text{ds}}$ output curves (inset) of the first-layer (**g**), second-layer (**h**) and third-layer (**i**) MoS₂ transistor.

Revision:

1. In page 12, line 251 of main manuscript, we revised the following sentence: “The transfer curves and output curves of the MoS₂ transistors of three different layers are shown in Fig. 4f–h, and the corresponding device images are shown in Supplementary Fig. 9. The devices in different vertical layers exhibit similar device performances (e.g., threshold voltage, on-state current), indicating our vdW vertical lamination process won't alter the intrinsic properties of bottom layer.”

2. In page 29, line 463 of Supplementary information, we added Supplementary Fig. 9.

REVIEWER COMMENTS

Reviewer #1 (Remarks to the Author):

Compared to the initial version and my concerns raised, I think the authors successfully made the needed corrections. Overall it is now an introduction of a very relevant MX2 transfer technique to vertical sidewall, and a comparison of it to planar MX2 integration that contains the needed detail to judge the potential scaling benefits. As it considers MX2 based devices, a yield and variability as can be obtained for Si based devices is implicitly not expected.

I think it is good for publication now.

more specifically:

1. I think they remain a bit too negative with respect to what is achievable with planar stacked integration and are deliberately doing a specific reference selection. The point that the used techniques are by nature only interacting with the wafer plane is somewhat oversimplified. I agree that a 60 layer stacked transistor is complex to produce, but it is not impossible considering what is done in stacked memory devices with gate vias that reach quite high layer counts today. However; as the main objective of the paper is propose an alternative MX2 integration scheme it is an acceptable oversimplification. I do not believe this will become a mainstream fabrication method for 2D based devices, but it might find specific applications.
2. With the nuances they explicitly added the skilled reader can make the assessment of the potential scaling benefit for the different applications. It is appreciated that in this context they mention the lower limit of the trench width and the fact that wafer level yield with many trenches has to be studied.
3. The area scaling is still bit optimistic as the contacts area that is horizontal, is not accounted for (but I agree this is application specific and hence hard to estimate), and they mention that limitation in the paper as well.

Reviewer #2 (Remarks to the Author):

The authors put a lot of effort into the revised manuscript, in particular the new configuration of device structure as well as improvement of device performance. The new estimation of the device density is more properly. Although I'm still not quite convinced with the estimated number provided, the interesting approach and data quality of the paper is suitable to be published in Nature Communication. I recommend publishing the paper in Nature Communication in the current form.

Reviewer #3 (Remarks to the Author):

I appreciate the author's great efforts in revising the manuscript. Most of my previous concerns have been resolved, except one key point that need to be clearly addressed before the final decision.

Response to my previous comment #6: I am grateful for the authors' supporting experiments in this regard, but the first layer still seems to be exclusively controlled by the top gate. Meanwhile, the other layers can be controlled by both the top and bottom gates simultaneously (see the inset illustration of Figure R12f). Can each MoS₂ device operate independently without being affected by the surrounding gate bias conditions? If the fabricated devices cannot be controlled intentionally, it would pose a significant problem for large-scale integration. More specifically, the second layer of MoS₂ can be capacitively coupled with the first layer gate metal, implying that the electrical performance of each layer would vary with the bias condition of the underlying metal gate.

Simply to confirm this issue, it is necessary to examine the threshold voltage shift of the second layer of MoS₂ as a function of the first layer gate bias, which would act as the bottom gate.

Furthermore, what is the Si substrate bias condition for this measurement? Is there any substrate bias dependency on the electrical performance for each layer?

Please elaborate on these issues clearly.

Responses to Reviewer #1:

General Comments: Compared to the initial version and my concerns raised, I think the authors successfully made the needed corrections. Overall it is now an introduction of a very relevant MX2 transfer technique to vertical sidewall, and a comparison of it to planar MX2 integration that contains the needed detail to judge the potential scaling benefits. As it considers MX2 based devices, a yield and variability as can be obtained for Si based devices is implicitly not expected. I think it is good for publication now. More specifically:

1. I think they remain a bit too negative with respect to what is achievable with planar stacked integration and are deliberately doing a specific reference selection. The point that the used techniques are by nature only interacting with the wafer plane is somewhat oversimplified. I agree that a 60 layer stacked transistor is complex to produce, but it is not impossible considering what is done in stacked memory devices with gate vias that reach quite high layer counts today. However, as the main objective of the paper is propose an alternative MX2 integration scheme it is an acceptable oversimplification. I do not believe this will become a mainstream fabrication method for 2D based devices, but it might find specific applications.

2. With the nuances they explicitly added the skilled reader can make the assessment of the potential scaling benefit for the different applications. It is appreciated that in this context they mention the lower limit of the trench width and the fact that wafer level yield with many trenches has to be studied.

3. The area scaling is still bit optimistic as the contacts area that is horizontal, is not accounted for (but I agree this is application specific and hence hard to estimate), and they mention that limitation in the paper as well.

Response: We thank the reviewer for the positive comments and support for its publication. In particular, we thank the reviewer to recognize and further emphasize the three important points (motivation, trench width, and scaling limitation) from our revised manuscript.

Responses to Reviewer #2:

Comments: The authors put a lot of effort into the revised manuscript, in particular the new configuration of device structure as well as improvement of device performance. The new estimation of the device density is more properly. Although I'm still not quite convinced with the estimated number provided, the interesting approach and data quality of the paper is suitable to be published in Nature Communication. I recommend publishing the paper in Nature Communication in the current form.

Response: We thank the reviewer for the positive comments and support for its publication in current form.

Responses to Reviewer #3:

General Comments: I appreciate the author's great efforts in revising the manuscript. Most of my previous concerns have been resolved, except one key point that need to be clearly addressed before the final decision.

Response: We thank reviewer for the positive comments. We also appreciate the remaining question (regarding to the electrical coupling) and would like to take this opportunity to further clarify below.

Specific comment 1: Response to my previous comment #6: I am grateful for the authors' supporting experiments in this regard, but the first layer still seems to be exclusively controlled by the top gate. Meanwhile, the other layers can be controlled by both the top and bottom gates simultaneously (see the inset illustration of Figure R12f). Can each MoS₂ device operate independently without being affected by the surrounding gate bias conditions? If the fabricated devices cannot be controlled intentionally, it would pose a significant problem for large-scale integration. More specifically, the second layer of MoS₂ can be capacitively coupled with the first layer gate metal, implying that the electrical performance of each layer would vary with the bias condition of the underlying metal gate. Simply to confirm this issue, it is necessary to examine the threshold voltage shift of the second layer of MoS₂ as a function of the first layer gate bias, which would act as the bottom gate.

Response: We thank the reviewer for these insightful questions. We fully agree with the reviewer that the device in previous Fig. R12f (also shown in Fig. R1a below) is actually a dual-gate transistor, and the second layer could be impacted by the gate voltage of first layer (as a bottom gate). However, there is a 1 μm thick PMMA layer between the first layer and second layer devices, serving as the low-κ interlayer dielectric. This layer has much smaller capacitance compared to the top-gate dielectric (~15 nm thick BN), hence could minimize electrical coupling between different layers, as highlighted in Fig. R1a.

Furthermore, following the reviewer's suggestion, we have measured the electrical properties of the second layer MoS₂ transistor (top-gated device) under various gate voltage of first layer (termed as V_{bg}), and extracted the threshold voltage (V_{th}) shift to examine the impact from interlayer coupling. As shown in Fig. R1b, c, the device in second layer exhibits consistent electrical performances with different V_{bg} from first layer, and its V_{th} remains stable under various V_{bg} , indicating the interlayer PMMA could effectively reduce the interlayer electrical coupling. Therefore, each MoS₂ device is mainly controlled by its own top gate and can operate independently without being affected by the surrounding gate bias conditions.

We thank the reviewer for these suggestions, and we have further explained the interlayer coupling effect in the revised manuscript.

Fig. R1. Dual-gate measurement of second layer MoS₂ transistor. **a**, Schematic of a dual-gate MoS₂ transistor. **b**, $I_{ds} - V_{tg}$ transfer curves of the second layer MoS₂ transistor under different V_{bg} from -5 V to 5 V, at V_{ds} of 0.1 V. **c**, The extracted threshold voltage as a function of V_{bg} .

Revision:

1. In page 12, line 257 of main manuscript, we added the following sentence: “ Furthermore, to avoid electrical coupling, PMMA interlayer dielectric ($\sim 1 \mu\text{m}$ thick) is also laminated between each layer. As shown in Supplementary Fig. 9e–g, the device in second layer exhibits consistent electrical performances with different gate voltages from first layer, and its V_{th} remains stable under various V_{bg} , indicating the low- κ PMMA layer could effectively reduce the interlayer electrical coupling.

2. In page 29, line 481 of Supplementary information, we updated Supplementary Fig. 9.

Specific comment 2: Furthermore, what is the Si substrate bias condition for this measurement? Is there any substrate bias dependency on the electrical performance for each layer? Please elaborate on these issues clearly.

Response: We thank the reviewer for raising up these questions. To ensure that each layer of the device has the same gate conditions, the Si substrate is floated (disconnected without applying bias voltage). We note the Si substrate is covered with a thick dielectric layer (300 nm SiO₂ and 500 nm bottom PMMA, shown in Fig. R1a above), which have a much smaller capacitance compared to the device top-gate dielectric (~ 15 nm BN). Therefore, electrical performance of each layer device is depended on by their top gates bias and largely unaffected by the Si substrate bias.

We thank the reviewer for these questions, and we have further clarified the Si substrate bias condition and its impact in the revised manuscript.

Revision:

In page 16, line 329 of main manuscript, we added the following sentence: “Note that the Si substrate is disconnected without applying voltage in the multilayer measurement. In addition, because the Si substrate is covered with a thick dielectric layer, electrical performance for each layer is largely unaffected by the Si substrate.”

REVIEWERS' COMMENTS

Reviewer #3 (Remarks to the Author):

I appreciate the author's additional efforts to fully resolve my previous concern, and the manuscript seems ready for publication as it is.